# Modern Kiln Burner Technology in the Current Energy Climate: Pushing the Limits of Alternative Fuel Substitution

**Maria Margarida Mateus** [1,2,3,*] **, Teresa Neuparth** [2] **and Duarte Morais Cecílio** [2,3]

1 Departamento de Engenharia Química, Instituto Superior Técnico, Universidade de Lisboa, 1349-017 Lisboa, Portugal
2 CERENA—Centro de Recursos Naturais e Ambiente, Departamento de Engenharia Química, Instituto Superior Técnico, Universidade de Lisboa, 1349-017 Lisboa, Portugal
3 Secil S.A., Fábrica Secil—Outão, 2901-182 Setúbal, Portugal
* Correspondence: margarida.mateus@tecnico.ulisboa.pt or margarida.mateus@secil.pt

**Abstract:** The current manuscript presents a review on existing kiln burner technologies for the cement production process, in the context of the current climate of energy transition and environmental remediation. Environmental legislation has become ever stricter in response to global climate change, and cement plants need to adapt to this new reality in order to remain competitive in the market and ensure their longevity. The cement production process is a well-established technology with more than a century of existence. There are several plants in operation whose process is outdated by modern standards, particularly considering the current industry decarbonization needs. The cement process requires tremendous amounts of energy, mainly recovered from the combustion of solid, liquid or gaseous fuels, which yields massive emissions of greenhouse gases. Thus, an important onus is placed upon the minimization of pollutant emission in the combustion system, as well as a substitution of fossil fuels with more sustainable alternatives. One of the sustainable alternative fuels comes in the form of refuse derived fuels (RDF). These high caloric fractions of municipal solid waste present a double advantage by reducing the amount of fossil fuels used and reducing the landfilling fraction of waste. However, their use in rotary kiln burners comes with important limitations for burner operation, namely that a high degree of control over primary air supply is needed to ensure complete combustion with minimal pollutant emission.

**Keywords:** cement; alternative fuels; burner; sustainability; energy transition

## 1. Introduction

The cement industry is one of the most relevant industries worldwide, as cement is the second most widely consumed product by mankind after water. Cement is the main raw material for concrete—the most widely used substance on the planet and a crucial material for construction and building industries. Although cement or concrete are the foundation of modern development, their manufacturing process is accompanied by significant pollutant emissions. Thus, the cement industry is one of the major contributors of greenhouse gases (GHG) emissions, such as carbon dioxide $CO_2$, $NO_x$, $SO_2$, Volatile Organic Compounds (VOCs) and others. Carbon dioxide emissions are, by far, the highest amongst all GHG emissions in cement production, making the cement industry accountable for approximately 7% of total $CO_2$ emissions generated by human activities worldwide [1,2].

$SO_2$ is generated from the sulphur compounds in the ores and the combusted fuel. Likewise, nitrogen oxides are generated through combustion of fuel in rotary cement kilns from the nitrogen in the fuel and incoming combustion air. In addition, incomplete combustion of fuels of many types is also an important source of VOC discharge to the atmosphere. Finally, $CO_2$ has two sources in the cement production process, which take place in the calciner and kiln burner equipment. In the pyroprocessing system, described in the following section, calcination reactions take place in order to form $CaCO_3$, an important

intermediate in cement production, and it also produces $CO_2$ as a secondary product. The second source of $CO_2$ is also located in the pyroprocessing system, in particular in the kiln burner that burns fuels of various types in order to provide the necessary heat, crucial for clinker production reactions in the rotary kiln [1].

GHG emissions are originated from different points of the cement production process. Calcination reactions account for approximately 50% of GHG emissions; 40% is related to fuel consumption and the remainder results from indirect emissions related to electricity consumption and vehicle traffic [2]. The 2050 roadmap of the European Cement Association CEMBUREAU aims for total carbon neutrality across the entire value chain, but concrete steps need to be taken to achieve said goal. To that end, the 2030 roadmap aims for a maximum $CO_2$ emissions value of 472 kg $CO_2$/t of cement, down from an average value of 580 kg $CO_2$/t of cement in 2020 [3,4].

Current emission limits in the European Union for pollutants such as $NO_x$, $SO_x$ and particulates are set at 200–450 mg/m$^3$ for NOx, 50–400 mg/m$^3$ for $SO_x$ and approximately 20 mg/m$^3$ for particulates (data from Germany and Austria) [5].

Because a large proportion of GHG emissions are derived from fuel combustion in the kiln burner, it is of utmost importance to explore the technologies and pathways related to this equipment that can help reduce emissions and improve process sustainability while guaranteeing its economic viability.

The kiln burner is an integral component of a rotary kiln system, part of the pyroprocessing system, serving to optimize the combustion of fuels and heat release in the kiln. The burner system is used for the sole purpose of providing heat to the rotary kiln to activate the chemical reactions that yield cement clinker [6].

In its early stages, burner systems simply consisted of the burning of non-processed fuels inside static kilns in lump form. Throughout the years, the cement manufacturing process has developed, resulting in the rotary kiln system, which includes the kiln burner system. More sophisticated technologies were required to prepare and inject fuel into the kiln system with complementary equipment, adapted to the nature of the type of fuels used. The development of these technologies and designs were motivated not only by environmental legal restrictions but also by productivity improvement goals.

Nowadays, the current landscape in modern cement production is focused on the mitigation of GHG emissions. To that effect, one of the first steps is the substitution of any fossil fuels used in the kiln burner with alternative fuels.

These alternative fuels, while still not solving the problem of GHG emissions, do assist in promoting a sustainable process integrated into the concept of a circular economy. Refuse derived fuels (RDF) are a major player in the cement industry currently, particularly in Europe and Asia. These help in providing a use for waste materials such as municipal solid waste and, after a pre-treatment, a high calorific value fraction can be separated and employed as fuel.

Due to the extremely high amounts of waste produced today by human populations, the volume of RDF produced from waste can be considered as a primary choice for alternative fuel in cement kiln burners. Alternative fuel substitution rates of up to 85% can be achieved, without resorting to other alternative fuels, thus allowing for the possibility of zero fossil fuels, without extreme efforts in the part of technological conversion of pre-existing lines.

However, when considering 100% alternative fuels, an important issue arises for an operating cement production plant. Due to the considerable age of the average cement plant's process, the currently installed burners will probably not be adapted to this complete substitution just yet, particularly when solid fuels are concerned. The granulometry of coal fuels and their composition is well known, easily controllable and, as such, allows for an easier operation of the kiln burner.

The inherent strong variations associated with the physical and chemical properties of RDF constitute a major challenge to be overcome in burner design. To that effect, we herein

provide a review on current burner technologies aimed at securing a smooth and productive transition towards complete alternative fuel substitution in the current landscape.

This work focuses on presenting and comparing modern kiln burner technologies in the current energy climate, as pertaining to maximizing alternative fuel processing and minimizing environmental impact. To the best of the authors' knowledge, there are no recent reviews published on the subject.

## 2. The Pyrosystem

We begin with a short overview of the pyrosystem. The pyrosystem is made up of the preheater, the kiln system and the clinker cooler [7]. The raw material enters the preheater, which usually consists of a cyclone tower and a calciner, where part of the fuel is burned in order to reach even higher temperatures, which allows for the majority of calcination reactions to occur. At the end of the pre-heater, the material reaches a temperature of circa 850 °C [8]. Afterwards, the material continues into the rotary kiln, where it continues to be subjected to heating from the flame of the clinker burner (at temperatures circa 1500 °C), which completes the clinkerization reactions. Lastly, the clinker passes through the air cooler. An air stream passes through the cooler and rotary kiln, thus removing heat from the clinker and the flame, and later passing through the preheater to serve as a heat source for the raw material [9].

The kiln burner is a vital piece of equipment in the pyrosystem, which is responsible for clinker formation. The kiln burner passes through the kiln hood, at the hot outlet of the kiln, as shown in Figure 1, and through which the fuel is transported across into the kiln with the help of the primary air stream at high velocities. The air stream passes through the burner pipe, carrying the fuel and conferring momentum. The primary air can also be used as axial and swirl air, which may be used to control the flame shape. In short, the kiln burner is perhaps the most crucial piece of equipment, responsible for the production of the thermal energy required to induce calcination or clinkerization. Incremental optimizations implemented in a burner system can lead to profound effects noticeable throughout the entire clinker production process [10].

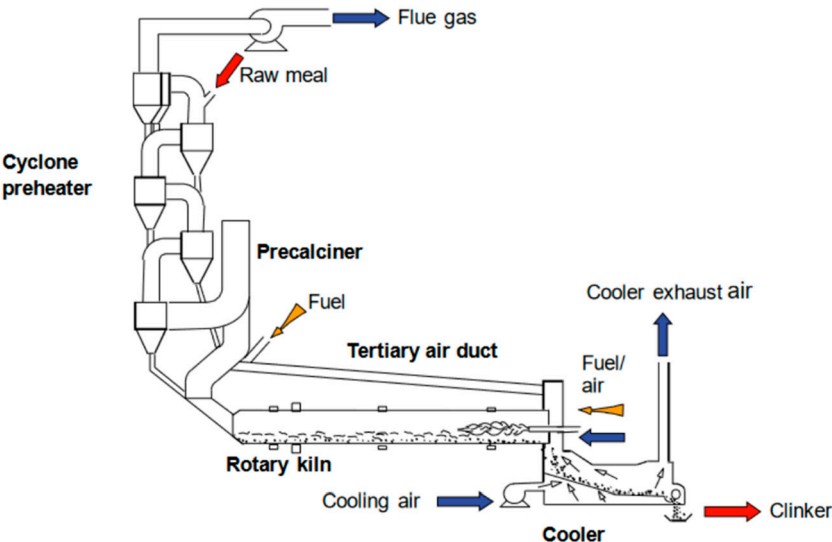

**Figure 1.** Rotary cement kiln (five-stage cyclone suspension preheater plus calciner plus high-efficiency cooler) [10].

The cooling air is used to cool the newly formed clinker in the cooler, thereby exchanging heat with the clinker, and it is then directed into the kiln, outside the firing pipe, resulting in what is referred to as secondary air. The secondary air provides the bulk of oxygen supply needed for combustion, making up 85–95% of the air required to combust the fuel [11]. It is essential that the amount of combustion air be accurately estimated, so as

to achieve a high thermal efficiency inside the kiln. If the amount of combustion air is too low, combustion will not be complete and CO emissions will occur, as well as emissions of unburnt fuel.

Some factors that improve combustion rates are increased primary air velocity and swirl (primary air rotation) for a better mixing for fuel and air, as well as high secondary air temperature for the ignition of combustion reactions and high ratios of air/fuels.

Thus, the ideal kiln burner combines three factors, which are optimizing heat economy, ensuring clinker quality and user-friendliness in operation [11,12].

Heat economy optimization requires a short, narrow and highly radiant flame to provide efficient combustion with low primary air and transport air requirements. Additionally, it ideally leads to the minimum excess air required to reduce thermal loss. Burner design must also provide control over the temperature and shape of the flame, providing efficient mixing of the hot secondary air and minimizing fuel dropout.

Additionally, to ensure clinker quality requirements, the burner must avoid reduced burning conditions in the clinkerization zones in order to obtain the clinker crystalline structure and ensuring no permanent overburning.

Finally, in terms of user friendliness, the burner must provide flexibility in flame adjustment, allow for multiple fuel operation and allow for stable ignition and flame operation during startups, all while avoiding high wear rates.

There are several offerings from existing manufacturers that achieve most—if not all—of the above requirements. The issue is that they rely on a significant amount of fossil fuels to achieve stable operations. The current energy and climate landscape urges the cement industry to push the limits of alternative fuel substitution for burner operation. High alternative fuel rates introduce instability in the process, a result of the varying composition, granulometry and foreign elements in the fuels themselves. As such, we present an overview of the fuels used in kiln burners, followed by the issues involved with high alternative fuel rates (particularly relying on RDF) and a section on modern burner technology, followed lastly by a critical analysis of their adequacy in the near future of the industry.

## 3. Fuels

The selection of fuels to be used in the burner plays a relevant role in pyroprocessing. Fuels take up a significant part of the operational costs, which vary according to availability, fuel handling and maintenance. In addition, the selected fuel must adhere to national and/or international standards. If it does not present the required quality, it can impact clinker production and quality. Finally, as mentioned before, the burner accounts for 40% of $CO_2$ emissions [2], and also produces additional flue gases such as CO and NOx, which depend on fuel selection. Thus, the choice of fuel is of the utmost relevance in the results regarding the overall environmental impact of the plant.

### 3.1. Coal

Coal is a primary fossil fuel, solid carbon-rich material that can be used as a source of energy. Several types of coal exist, each classified based on carbon content. The types of coal used in the cement production processes, such as anthracite, bituminous coals, and lignite, vary according to region and availability, with bituminous coal the most commonly used.

The burning of fossil fuels, including coal, releases large quantities of $CO_2$ into the atmosphere and is a major driver of global warming [13].

Despite the high environmental impact, coal combustion is still responsible for 90% of the energy consumed by cement plants worldwide; because coal is more abundant and easily extracted compared to other fossil fuels, it has been typically used as an energy source in the cement industry.

However, in recent decades, coal usage has decreased due to an increasingly important drive for alternative fuels, which is based on environmental concerns over greenhouse gas emission (mainly $CO_2$) and issues related to waste management [14].

### 3.2. Petroleum Coke (Petcoke)

Petroleum coke is a by-product of petroleum coke cracking after refining, with high carbon content, high calorific value, low ash content and volatile characteristics [15]. In recent years, petroleum coke has become a major fuel in the cement industry, because the price is favourable compared to other fossil fuels. Although petcoke is preferred due to its price, it presents a high sulphur content, and it is usually less reactive than most coals. Its lower volatile content makes it difficult to ignite. This property requires a fast mixing of the fuel particles and secondary air so that enough oxygen if provided and, thus, ignition is achieved as fast as possible. This requirement influences burner design, because a higher momentum is desired, as well as a multichannel strategy [15].

The cement industry employs cost-effective fuels at the lower end of the quality scale, which applies to both coal and oil.

Usually, the oil used in the cement kiln is heavy oil (API No 6), which is the main liquid residue from refining processes such as hydrocracking and visbreaking. The oil is a mixture of complex aliphatic and polycyclic aromatics compounds (typically around 50:50), with negligible vapour pressure. Because sulphur compounds such as mercaptans tend to be non-volatile, the sulphur content is usually relatively high. Use of this oil requires it to be heated in order for it to be easily pumped, and nebulized at the burner to provide sufficient surface area for ignition to occur efficiently [16].

### 3.3. Waste Fuel

In the last two decades, the use of alternative fuels has increased exponentially. Waste fuels are derived from waste fractions (i.e., municipal solid waste) which are purified and separated in an attempt to isolate a fraction of the material possessing lower moisture content and higher calorific value to be used as fuel. One of the main reasons for this growing interest is the significantly lower cost when compared to fossil fuels. It has been estimated that a cement plant with an annual production of 1 million tons of cement could save up to EUR 2.4 million annually by replacing 30% of fossil fuels with no-cost alternative fuels [17]. In some cases, cement plants are incentivized to replace part of their fuels with alternative fuels, often in the form of subsidies.

Another major drive for the use of alternative fuels is the reduction of $CO_2$ emissions. Although significant differences can occur depending on the source of the fuels, overall, the use of alternative fuels results in much lower $CO_2$ emissions, because most of the fuels are partly biogenic and $CO_2$ neutral. Typical waste fuels used in fuel combustion in the cement industry are [18]:

- Refuse Derived Fuels (RDF);
- Landfill Gas;
- Tyre Derived Fuels (TDF);
- Meat and Bone Meal;
- Biomass Sludges;
- Waste Liquids.

The $CO_2$ reduction mechanism from using alternative fuels depends on the nature of the fuel itself. Biomass-based fuels are renewable energy sources, and waste derived fuels (RDF, TDF, meat and bone meal, etc.) are waste stream resources with significant energy content, thus reducing $CO_2$ emissions by avoiding the typical end-of-life disposing mechanisms and physical–chemical process that result in higher $CO_2$ emissions.

Furthermore, the use of waste fuels in the combustion process presents an additional advantage in the diversion of waste from landfills. Fuel ashes are incorporated into clinker, which eliminates the solid waste stream that is otherwise encountered during waste incineration. Although it may be beneficial to use alternative fuels, the utilization of waste fuels faces some challenges related to a larger particle size, higher moisture content and decreased combustion temperatures. Therefore, it is important to take into consideration all these aspects when choosing the type of fuel used in the combustion process.

### 3.4. RDF Operational Challenges

In this section, we will dive deeper into the issues involved with the processing of refuse derived fuels, one of the most widely used waste fuels in the cement industry. In a constant push towards decarbonization, complete replacement of fossil fuels is mandatory. RDF present themselves as a promising candidate.

## 4. Evolution of Kiln Burners

In the early 1900s, the kiln burner was modified from being mainly coal fired to being mainly oil fired. During the oil crisis of the 1970s, rising oil prices caused the type of fuel to be switched back to coal or petcoke. Since the 1990s a demand for an increased amount of alternative fuel firings has arisen, associated with environmental legislation that also became stricter, setting limits on the emission of pollutants such as $NO_x$ [19]. In order to adapt to the varying fuels and process requirements, the kiln burner has also undergone technological development. While the burners themselves have changed, there has also been an important change in burner operation, from direct firing to indirect firing, since the 1970s. The directly fired burner is connected directly to the fuel mill. While this is a simple system, it has the disadvantage of the primary air being used to dry the fuel before it is admitted to the kiln. This will result in larger amounts of primary air being used and unnecessary water in the kiln flue gas, which in turn results in lower thermal efficiency. It is also more difficult to regulate the system, because simultaneous control of the fuel feed and the mill speed is required. Additionally, mill failure will unavoidably lead to a kiln stop. In indirectly fired systems, an intermediate storage of the pulverized fuel is installed after the mill. This reduces the disadvantages of the direct fired system, but also requires additional equipment. Another advantage of the indirect fired system is that one mill can feed both the kiln and the calciner [11].

### 4.1. First Generation

First-generation kiln burners were introduced before the 1960s [12]. They were characterized by having a monochanel for fuels (coal and gas only), which meant that primary air and fuels were mixed and simultaneously injected in the same channel, as presented in Figure 2. Typically, these burners produced a long, slim flame with low radiation intensity. They were used in long kilns, in wet and semi-wet processes with no calciners and no preheaters. High primary air rates were necessary (15–45% of total stoichiometric air), and there was limited control of flame properties [12,19].

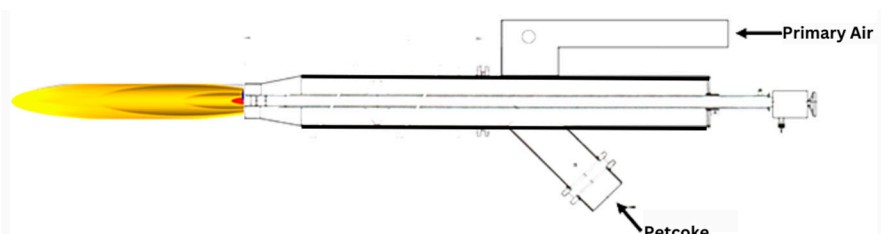

**Figure 2.** First-Generation burner.

### 4.2. Second Generation

Second-generation burners were introduced in 1980. Pillard, FLS Smith and KHD were some of the pioneer companies in second-generation manufacturing.

The burners were used in shorter kilns with a preheater and separate calciner (dry process). Unlike their first-generation counterparts, second-generation burners were designed so fuel was injected through two separate channels, as shown in Figure 3. A better control of flame shape was provided, and there were faster mixing rates when compared to the previous generation of burners. In addition, lower primary air rates were required, improving heat economy [12].

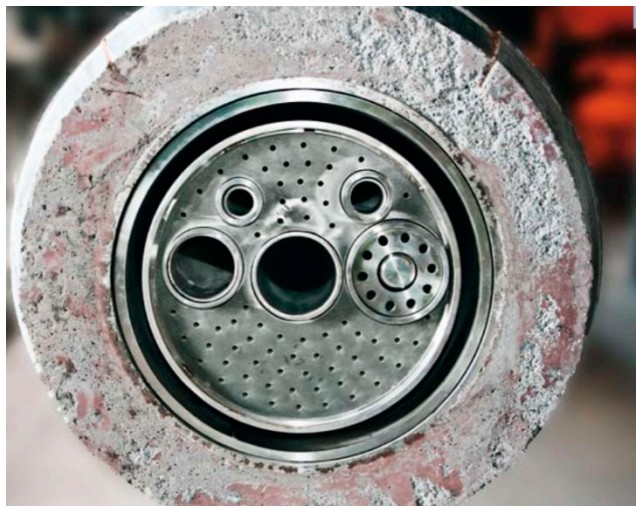

**Figure 3.** FLSmidth Duoflex G2 burner tip [20].

The main types of fuel used were coal and petcoke due to the oil crisis in the 1970s. In this decade, indirect firing was employed, which increased thermal efficiency.

This generation introduces swirl air that, as mentioned, improves fuel and air mixing, provides a higher flame control and gives higher momentum and, thus, improved clinker quality [11].

### 4.3. NO$_x$ Reductions

In cement production, NO$_x$ formation can have different sources, but it is typically formed at high temperatures from the reaction between the molecular nitrogen and oxygen present in the combustion air and nitrogen present in the fuel. Reductions of NO$_x$ emission can be achieved by [12]:

- reducing overall combustion temperature;
- avoiding short duration temperature peaks in the process;
- avoiding local temperature peaks in the flame;
- reducing oxygen concentration;
- reducing retention time of fuel in high oxygen environments.

In this the late 1980s, some environmental concerns regarding NO$_x$ emissions originated, and some changes were made in order to reduce these emissions. For instance, ignition close to the burner tip is preferred as it tends to lower the NO$_x$ emissions [10].

### 4.4. Third Generation

Third-generation kiln burners are considered to be the burners currently available. They are often personalized to the customer's needs but, in general, last generation kiln burners are momentum and flame optimized, release lower pollutant emissions, are multi-fuel compatible and use lower amounts of primary air. Currently, FLS, Pillard and KHD are some of the most relevant companies regarding the offering of kiln burners.

### 4.5. FLSmidth

The available technical specifications of the Duoflex G2 burner are as follows:

- Burner capacity: 20–250 MW;
- Solid, liquid and gaseous fuel support;
- Primary air pressure: 250–400 mbar;
- Primary air consumption: maximum 15% (standard 6–8%);
- Axial/radial air ratio adjustable (standard 1:2);
- Primary air momentum: 1250–1780% Primary air velocity.

The Duoflex G2 burner, developed by FLSmidth, Inc. in 2008, [20] is a multifuel burner as it can fire pulverised coal or coke, oil, natural gas or any mixture of these. It may be fitted with extra ducts for secondary or alternative fuels. The burner is capable of any combination of fuels, with a thermal capacity from 20 to 250 MW. Regarding the burner design (Figure 3), it features two concentric annular primary air ducts that form two primary air channels, one for axial air and one for radial air, that surround another annular duct for the firing of solid pulverized fuels such as coke and coal [21].

It can be adjusted for alternative fuels (Figure 4) and it presents two central ports for alternative fuels—a high-pressure duct for fine solids and another possibly offset pressure duct for rougher fuels, in order to change the trajectory and increase residence time of fuel in the flame.

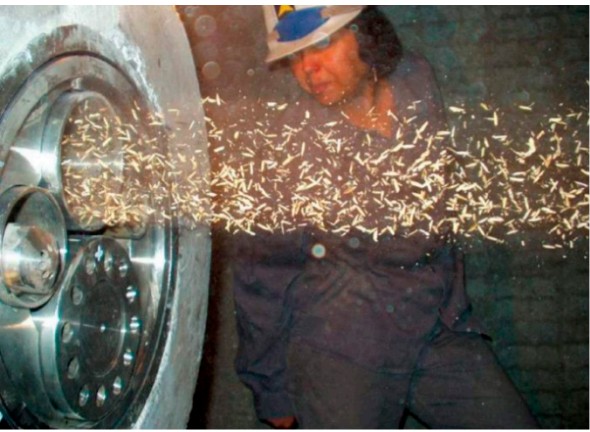

**Figure 4.** FLSmidth DUOFLEX G2 burner using waste fuels [21].

Two air flows (Figure 5) are mixed before being injected through a conical air nozzle. Because axial air and radial air share a common nozzle, the degree of swirl (axial/radial air ratio) can be altered without variations in momentum. Swirl is responsible for air recirculation, which stabilizes the flame and provides a short ignition. However, if there is too much swirl, it can provoke high temperatures caused by flame impingement. Therefore, it is important to have good control over swirling parameters. Primary air consumption typically varies between 6 and 8% (although it can reach a maximum of 15% in special circumstances), corresponding to a primary air momentum of 1250 to 1780% m/s [21].

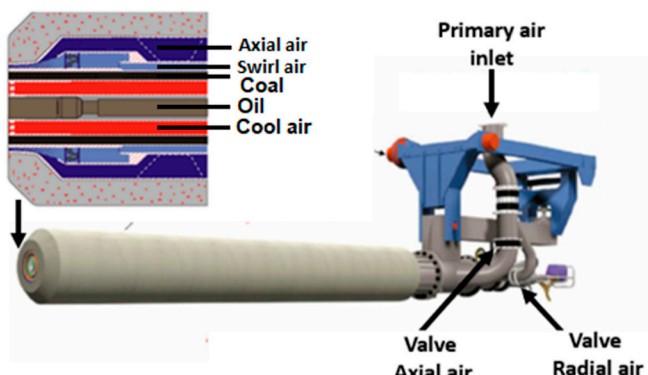

**Figure 5.** FLSmidth DUOFLEX G2 burner configuration [21].

The Jetflex burner available technical specifications are as follows:

- Maximum burner capacity: 250 MW;
- Flame momentum: 7 N/MW to 11 N/MW;
- Solid and liquid fuel support.

Jetflex by FLSmidth is the latest kiln burner technology of the company. The design is targeted towards traditional fuels such as oil, gas, pulverised solid fuels and high and medium-grade alternative fuel qualities and is available at capacities up to 250 MW. Regarding burner tip configuration, the burner can be configured in several ways depending on the fuel quality and type (Figure 6) [22].

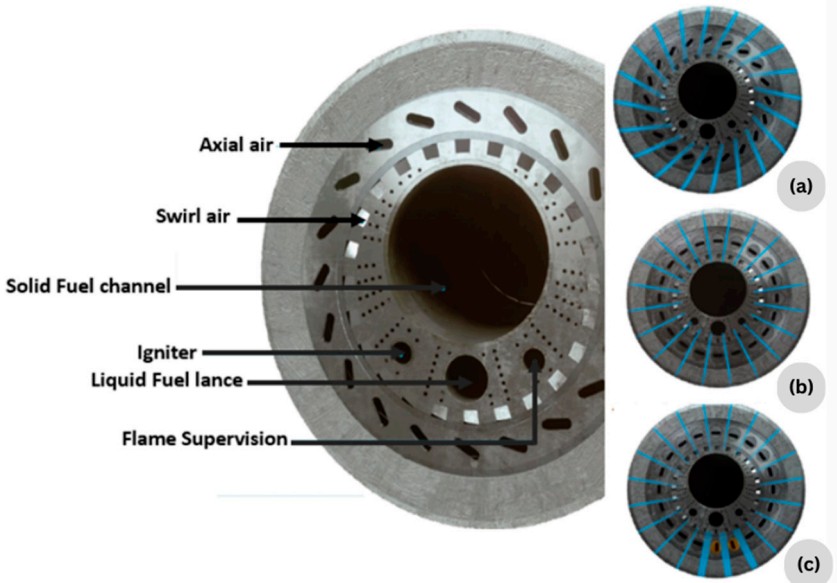

**Figure 6.** FLSmidth Jetflex burner tip. (**a**) Open configuration increases entrainment of secondary air; (**b**) Annular configuration makes it possible to shape the flame and is necessary for fuel stopping and lifting; (**c**) Fuel lift configuration increases air flows under the solid fuels, extending retention time [22].

The burner has a straight uninterrupted channel, wherein solid fuels (pulverized or alternative fuels) can flow undisturbed, which in turn reduces both blockages and wear. This improves heat and power consumption, decreasing cold airflow from fuel transport. The Jetflex has effective rectangular jet air nozzles and a swirler that allows for solid fuel flow through a single channel, instead of the traditional annular fuel channel for pulverized fuels. The jet air nozzles used for ejecting swirl air consist of several rectangular shaped nozzles with a fixed angle concentrically surrounding the fuel, which allows for a well-defined mixing of hot secondary air with the fired fuel [22].

Located on an external ring at the tip there are 20 nozzles for the axial air. The nozzles can turn individually 360° and further help to shape the flame.

The burner momentum and flame shape are controlled by regulating primary air pressure and flow. In general, the primary air rate may vary, enabling a flame momentum of 7 to 11 N/MW.

The Jetflex burner has two additional features compared to the standard Jetflex burner: rotatable jet air nozzles and a retractable centre pipe for alternative fuel firing.

The rotatable jet air nozzles (Figure 6) allow for a more efficient adjustment of the flame shape, according to fuel and process requirements.

The retractable centre pipe for alternate fuel firing enables significant decreases in fuel velocity at the burner outlet, which increases fuel retention time in the flame and allows for early ignition, which is suitable for low grade fuels. Better flame control, even with low grade fuels, contributes towards higher alternative fuel ratios in the burning process [22].

- Fives Pillard.

The Rotaflame burner available technical specifications are as follows:

- Burner capacity: 5–180 MW;
- Primary air fraction: 6–10%;

- Primary air pressure: 150–250 mbar;
- Solid, liquid and gaseous fuels support;

The Rotaflam burner is one of the Pillard's burners designed for the cement industry. It has large capacity for a variety of fuels such as coal, coke, oil, natural gas, solid alternative fuels, waste oils, solvents and process gases, with a substitution rate of up 80% and an output range of 5–180 MW [23,24].

Axial air and swirl air ducts are respectively located by the outer ring channels, followed by an annular channel for pulverized fuel. The latter surrounds pipelines designated for natural gas, oil and alternative fuels [24].

Around the solid waste channel there is a jacket pipe where a portion of primary air is blown into a patented swirler nozzle. In some cases, a jacket pipe is used to blow the waste fuel directly into the centre of the flame, which increases flame length and possibly causes fuel particles to leave the flame contour. In this case, the swirl nozzle causes the waste particles to follow a rotating trajectory, which leads to better fuel dispersion into the flame and higher residence time [25].

In general, primary air rates are 6–10% of total combustion air with a typical air pressure of 150–200 mbar and burner momentum of 8–12 N/MW [23,25].

As is the case of Pillard's Rotaflam burner, Pillard's Novaflam burner can employ a wide variety of fuels as it can use pulverized coal, petroleum coke, heavy and diesel oil, waste oils, solvents, natural gas and solid wastes, with a burner capacity from 10 MW to 200 MW [26,27].

However, unlike Rotaflam, the Novaflam burner has a single primary air channel (Figure 7) with two separate and controllable primary air outlets at the burner tip. The radial tip can be moved back and forth for flame adjustment. Typical values of primary air rate are between 7 and 10%, at 130 to 250 mbar pressure [28].

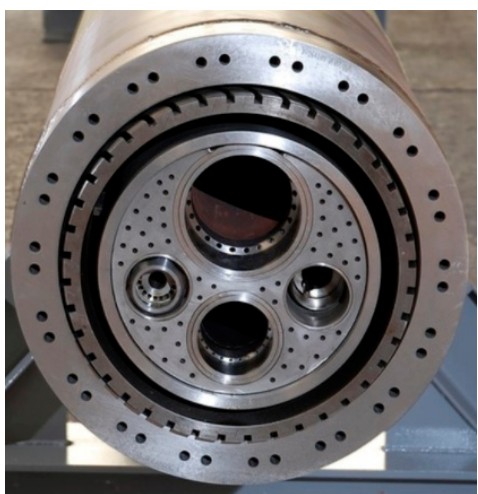

**Figure 7.** Five Pillard's Novaflam burner tip [28].

- KHD

Available technical specifications on KHD's Pyrojet burner are quite scarce, the only references being burner capacity varying from 10 MW to 500 MW and its support for standard fuels as well as various alternative fuels.

KHD has developed Pyrojet, which is a typical multi-channel burner for a variety of fuels. It contains axial air exits which leave the burner tip at high velocity through a set of jet nozzles located on the perimeter of the burner tip (Figure 8). This arrangement reduces the required amount of primary air in general and allows lower fuel consumption. The required primary air is 7–11% of total combustion air [29,30].

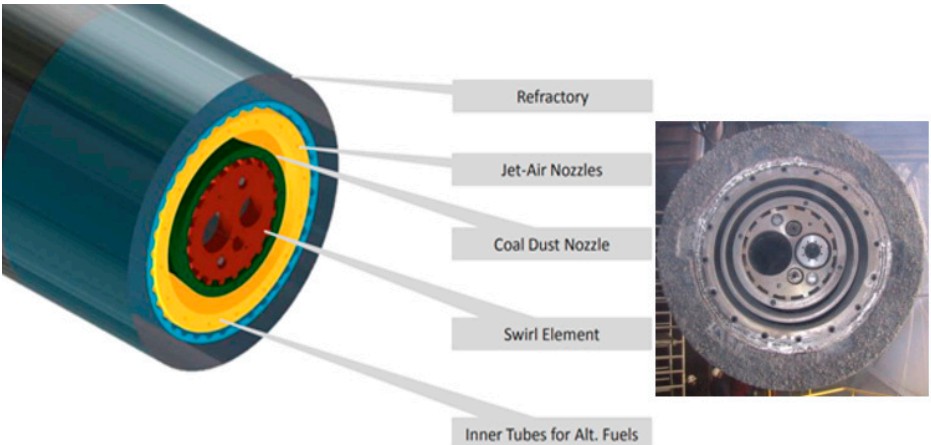

**Figure 8.** KHD Pyrojet burner tip configuration with retractable AF pipe [29].

Pyrostream is one of the best performing burners of KHD. It is designed to allow high alternative fuel substitution rates, although it is also suitable for mixed fuels.

The burner features 12 jet primary air nozzles at the burner tip, indicated in green in Figure 8, which are positioned outside the coal annular channel (in red), and at the centre (in yellow) is the main swirl element that consists of an alternative fuel channel and main swirl air channels. The jet nozzles are adjustable, individually or synchronized, and can be rotated 360°, either converging or diverging from the main swirling primary air flow. These adjustments allow for a more precise flame setting and shape (Figure 9). The burner design features low primary air amounts of 1.6% jet air, 2.4–4.4% swirl air and 1% cooling air (5–7%), resulting in a flame momentum of 4.5–5.5 N/MW (it can rise up to 11%, depending on customer requirements) [31].

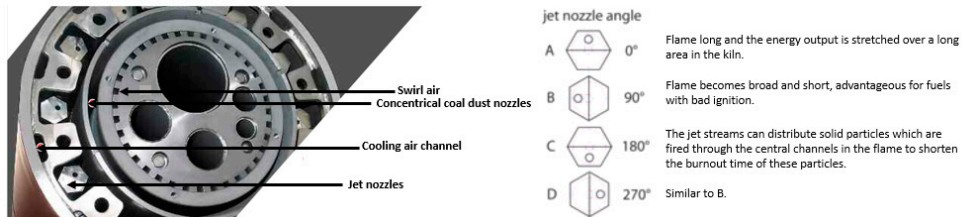

**Figure 9.** KDH Pyrostream burner tip and jet nozzles configuration [31].

- ThyssenKrupp

Availability of technical specifications for the POLFLAME burner is quite scarce, with the only reported parameter being a burner capacity of up to 300 MW for kiln production capacities of up to 12,000 t/day and can reportedly achieve complete alternative fuel substitution.

The POLFLAME burner from ThyssenKrupp is equipped towards employing traditional fuels such as fossil fuels, oil and gas or for the use of solid and liquid alternative fuels, achieving high fuel versatility. It presents an adjustable supply of oxygen to the heart of the flame, which assures good ignition and burnout of different fuels. Flame shaping is also controllable by adjustments of the nozzle and the end of the burner tip [31].

The burner has several primary air jet channels positioned around the central tubes that can be adjusted in radial or tangential direction, thus providing greater control over the flame. Around the primary air jets, there is an annular channel for pulverized fuel, and channels for oil and alternative fuels are located at the centre (Figure 10) [32].

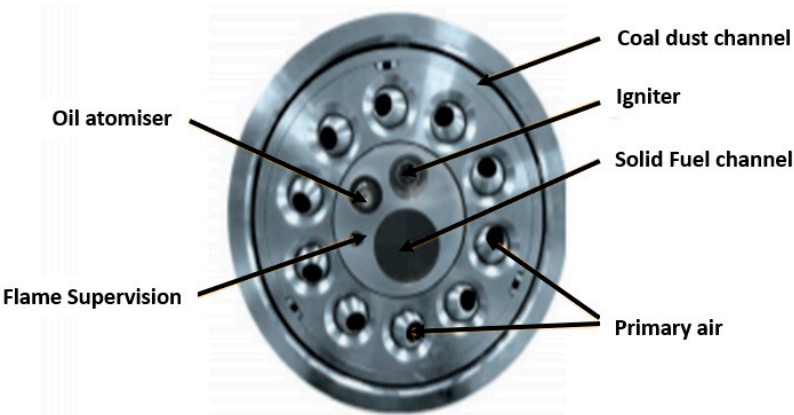

**Figure 10.** ThyssenKrupp Polflame burner tip [32].

- Greco

    The Flexiflame burner available technical specifications are as follows:
- Burner capacity: up to 175 MW;
- Primary air percentage: 10–13% of primary air;
- Primary air pressure: 50–350 mbar;
- Burner momentum: 6–7.7 N/MW.

    Flexiflame from GRECO (Figure 11) presents itself as a robust and flexible burner with a significant degree of control over the relative amount of primary air between the swirl and the dispersion channels, which offers adjustment between lower $NO_x$ emissions (higher swirl air ratios) and higher solid alternative fuel (more dispersion air). It has three main primary air channels, which are axial, radial/swirl and dispersion, with the fossil solid fuel channel located between the axial and swirl channels and the dispersion channel. In the centre, the solid and liquid alternative fuel ports are separately located.

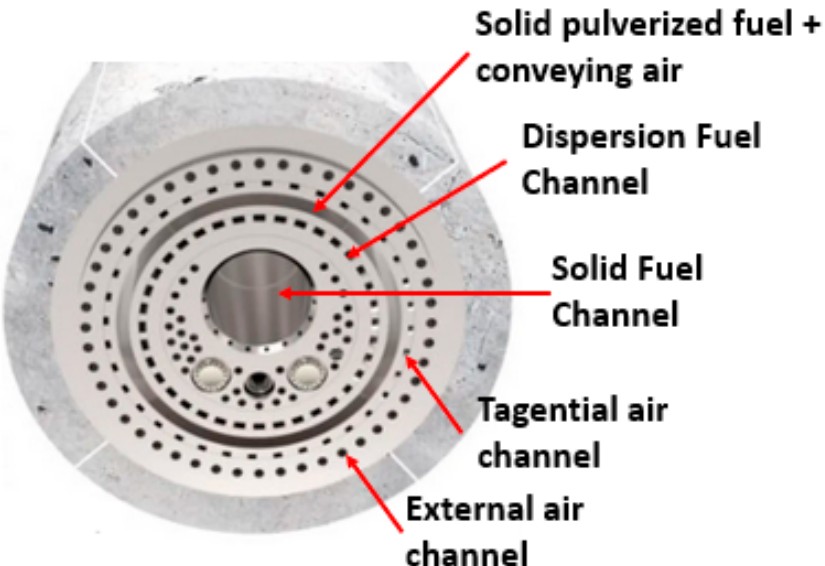

**Figure 11.** Greco Flexiflame burner tip [33].

    The primary air fraction varies between 10 and 13% of the required combustion air, of which 4–6% is external air, and tangential and dispersion air are both 1.5–3% of the stoichiometric air amount, with pressures of 200–500 mbar and a burner momentum from primary air between 6 and 7.7 N/MW [30,33].

- FCT

The available technical specifications for FCT's Turbu (Figure 12)-jet are as follows:

○ Burner capacity: up to 160 MW;
○ Solid, liquid and gaseous fuel support;
○ Primary air percentage: approximately 11% (based on a case study);
○ Can achieve alternative fuel substitution rates of 70% (based on a case study).

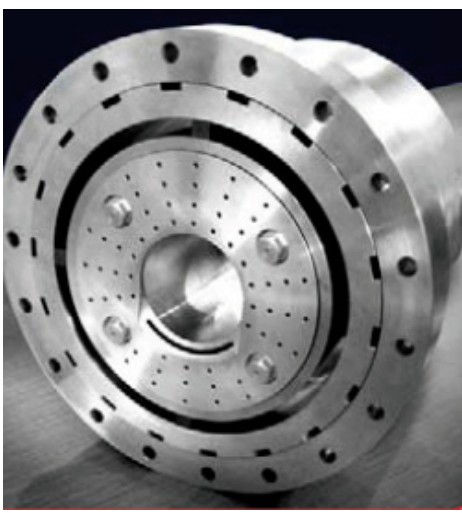

**Figure 12.** FCT Turbu-jet burner tip [34].

FCT's Turbu-jet, similarly to the present generation's available burners, is fuel flexible. It can be used in combination with all fuel, particularly with alternative fuels, regardless of their physical state, although its primary fuel is from fossil origin.

In this case, the flame shape is controlled by a single valve that alters the amount of primary air through the axial and swirl air channels. The typical primary air rates vary between 6 and 10%, with a pressure of 250 to 700 mbar [34]. The swirl air duct is incorporated into the pulverized fuel annular channel so as to form a better dispersion of the fuel cloud, which results in better mixing, and fuel ignites closer to the burner tip.

Usually, it presents a high momentum range of 6 to 14 N/MW, which provides a firm and steady flame, and has a burner capacity of from 10 up to 160 MW [30,34,35].

● Dynamis

The available technical specifications of Dynamis' D-flame burner are as follows (based on a case study):

○ Primary air percentage: 11.3%;
○ Primary air pressure: 120–430 mbar;
○ Burner momentum: 8.1 N/MW.

Dymanis' D-flame presents multifuel capacity, with a rate of substitution of fossil fuels by waste derived fuels above 40%.

The D-flame burner has three shaping air streams. (Figure 13). One external air stream is axially injected into the kiln and impacts the turbulence index and consequently is responsible for the mixing of the secondary air with the flame. It has an air stream that can be injected into the kiln both tangentially and axially. This stream aims to increase stability through internal recirculation of air, subsequently conforming the flame. Finally, the burner has an internal air stream that is axially injected into the kiln, ensuring the proper dispersion of solid fuel particles. The arrangement of these three air streams changes the flame shape [35].

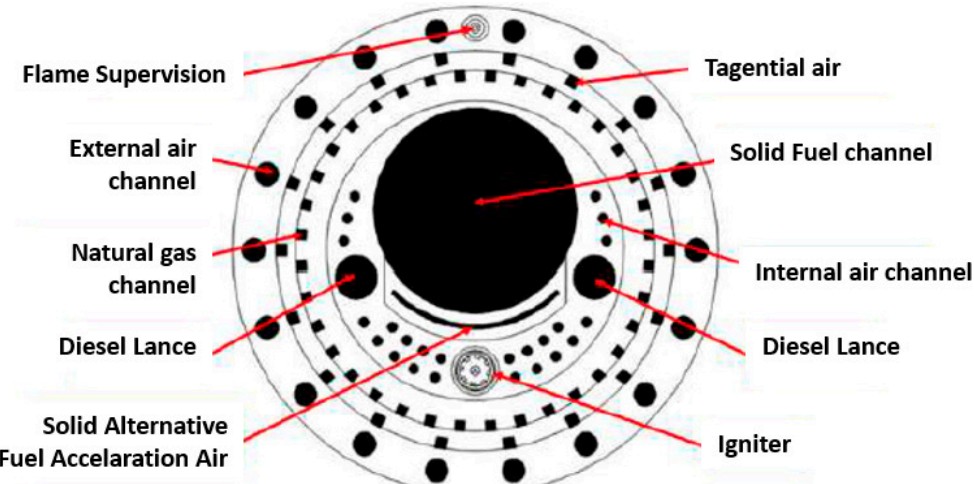

**Figure 13.** Dynamis D-flame burner tip [35].

## 5. Discussion

Kiln burner generations are defined by different operational paradigms. The first generation of burners were used in wet or semi-wet production processes, where water was used to homogenize the raw material. This meant that a significant portion of the fuels were burned to evaporate water, rather than simply processing the raw material.

The second generation of burners was introduced in dry processes with shorter kilns. They focused on improving fuel injection (multi-channel injection) and primary air insertion to provide better flame control, employing mostly oil and coal as fuels.

Modern kiln burner technology, however, is focused on processing alternative fuels and maximizing their insertion, as well as improving the energy efficiency of the combustion process.

It is difficult to objectively assess burner performance. It is of the utmost importance to stress the fact that a great number of the resulting burner metrics (including emissions) are a direct result of the capacity of the rotary kiln where they are installed. All main kiln burner manufacturers are focused on improving the equipment, allowing for the use of alternative fuels while maintaining effective combustion and reducing GHG emissions. Through their proprietary technologies, the information that is available to the general public is at times lacking and therefore insufficient to establish a complete comparison. However, we do believe that enough information is available to achieve significant conclusions.

Despite no specific operating requirements provided by the burner manufacturers, it is generally accepted that a high momentum, low primary air rates, high alternative fuel substitution rates, an effective mixture between primary air and secondary air and low GHG emissions are required. However, as observed in a research project by VDZ [1], the recommended minimum axial momentum for alternative fuel firing is 6.5 N/MW and a swirl no. from 0.01 to 0.12, depending on burner type and based on industrial cement kiln trials of refuse derived fuel (RDF) firing with substitution degrees of up to 60%.

Another crucial aspect to consider is $NO_x$ formation. Thermal and fuel $NO_x$ usually forms in specific areas inside the flame where peak temperatures occur, as well as where there is a high oxygen concentration required for higher burnout rates. $NO_x$ formation is a relatively common occurrence during the firing of alternative fuels, because these require higher temperatures to ignite, due to lower heating value when compared to fossil fuels. Therefore, a compromise must be achieved between high substitution rates of alternative fuels and low $NO_x$ formation, which also contributes towards the importance of primary air injection in the design of kiln burners [19].

Primary air is used to shape the flame and control the mixing pattern of combustion air, which is achieved through the high momentum, swirling or divergent air flows.

A common feature for primary air injection systems are primary air jets, which may be located on the inside or on the outside, or in the annular pulverized fuel channel (if present on the burner). If the burner has the primary air jets positioned on the inside of this channel, like the Polflame, it may achieve higher peak flames near the burner zone, although a fast decrease in downstream kiln temperature is to be expected. The higher peak temperatures may result in higher $NO_x$ emissions when compared with burners presenting primary air jet located on the outside of the annular pulverized fuel channel, as is the case of the Rotaflam model, due to a better mixing of the fuel with the swirl air, resulting in generally wider and shorter flames. However, this phenomenon may be reduced by increasing burner momentum.

High burner momentum means the formation of short, stable flames, whereas lower momentum generates longer, weaker flames. General guidelines dictate a burner momentum of approximately 13 N/MW [35], and most burner models meet the value. Despite this specific guideline, some burners can present lower momentum and still achieve desirable kiln burning efficiencies and operating conditions. A lower burner momentum can help achieve a lower flame temperature that limits the thermal $NO_x$ formation mechanisms and thus results in lower $NO_x$ emissions. In fact, modern burners present a trend moving away from higher burner momentums to avoid the formation of extremely hot flames.

Additionally, it is practically impossible to completely prevent the formation of $NO_x$, which creates the need for cleaning technologies. Most—if not all—cement plants present a scrubber where the exhaust gas is cooled down and an ammonia solution is introduced, in order to reduce the $NO_x$ emissions to acceptable levels.

Another important feature is fuel flexibility, because the use of alternative fuels, especially solid alternative fuels, is rising due to environmental concerns related to the use of fossil fuels. Although most burners, like the ones mentioned in this article, are equipped to accommodate 100% solid alternative fuels, the actual implementation is quite complex.

Solid AF particles are typically larger than pulverized fossil fuels particles, which adds difficulty in ensuring a good mixture with the primary air, which in turn leads to delayed and ultimately incomplete combustion. Additionally, higher air flowrates are required to keep the fuel particles suspended. If the particle transport fails, the denser particles may fall out of the burner and partially burn on the clinker bed, which creates an undesirable "brown" clinker and decreases clinker quality [35]. As an attempt to mitigate this factor, a few parameters can be considered. For instance, the ratio between the burner's outer diameter and the internal kiln diameter must be 12–15%, as for higher ratios the swirl air may not properly mix with all fuel particles.

Other alternatives include primary air channels that can be installed adjacent to the AF pipes. Flexiflame installed air jets around the alternative fuel pipes, while the dispersion air is ejected at the burner centre. FLS Jetflex has a retractable AF pipe that creates a small expansion chamber which, combined with the axial air nozzles, decreases the velocity at the burner tip and consequently increases fuel residence time in the flame, enabling early ignition. However, the overuse of primary air may have consequences as high primary air rates decrease thermal efficiency by admitting greater amounts of cold air. Therefore, low primary air rates of about 10% are advised [35]. It is also important to take into account other parameters besides burner design, such as secondary air temperature and burner position inside the kiln.

In this current environmental and transitional landscape, the severe energetic requirements of cement production plants dictate that the burner system must be thoroughly optimized. This is particularly urgent for plants located in countries with severely restrictive environmental laws regarding the discharge of pollutants into the atmosphere, as is the case of European countries. As was previously mentioned, the substitution of fossil fuels with alternative fuels creates the added complexity for the burner to be able to deal with varying distribution, poor calorimetric performance and the added pollutant load that originates from alternative fuels with varying chemical compositions and relatively high impurity loads, as is the case of RDF.

Taking this into account, the main properties to optimize are the combustion efficiency in the pyrosystem and the minimization of pollutant emission, which often are contradictory. This means that an order of priorities for burner behaviour must be established.

Let us take into consideration an average cement production plant in Europe, which has to deal with strict environmental regulation. Its first and foremost priority is to substitute the majority (if possible, the totality) of fossil fuels with alternatives. While all the commercial burner solutions advertise the capacity to withstand 100% alternative fuel rate, this creates a need for higher flame temperatures in order to achieve complete combustion, leading then to higher $NO_x$ emissions due to thermal $NO_x$ formation mechanisms as well as fuel $NO_x$. The currently available burner technologies present three different mechanisms to approach this issue: higher primary air flowrates, an expansion chamber at the end of the burner, and jet air channels around the alternative fuel feed nozzle. The first two options can lead to operation limitations due to the maximum capacity of the primary air blowers, as well as the higher wearing rate created by the attrition between the alternative fuel particles and the tubing, and the lower thermal efficiency associated with a higher cold air flowrate. In comparison, the jet air nozzles around the feed are able to ensure a better mixing of the air and fuel particles with minimal impact to blower operation and a more complete combustion at lower temperatures. The Flexiflame and Novaflam burners present this feature, which may constitute an advantage for plants with these limitations.

The solutions that can help the cement plants reach their decarbonization in the long term are well agreed upon and understood. Technologies such as carbon capture, storage, and utilization are the most promising candidates as they are able to reduce not only fuel emissions but also process emissions resulting from the calcination of the raw materials. However, these technologies still require further maturation before their successful application at an industrial scale. The careful selection and optimization of the burning system according to available technologies is crucial to ensure a safe and efficient operation of the cement production process in current days, while simultaneously allowing for optimal alternative fuel substitution.

## 6. Conclusions

Cement production faces a difficult challenge in reducing greenhouse gas emissions. Efforts have been made in finding a solution to this problem, particularly in the kiln burner with the use of alternative fuels. It is not possible to determine which of the available kiln burners is the best. All latest kiln burner technologies present options for the usage of alternative fuels, especially because this implementation is complex.

Although it presents lower costs, reduces waste and reduces $CO_2$ emissions, using alternative fuels has its disadvantages, such as higher $NO_x$ emissions and less stable flames due to heavier and larger particles that may cause temperature peaks and falls in the clinker bed. Burner manufacturers have been finding solutions for these problems by adjusting primary air flows and pressures, diversifying primary air outlets in number, adjustability and angles and changing positions of fuel pipes. Oxygen enrichment and co-firing may also be potentialy beneficial regarding the use of alternative fuels. All these features help to increase flame controllability and improve combustion efficiency and heat economy, thus achieving necessary clinker quality. Additionally, there must be a compromise between fuel substitution rate and low $NO_x$ emission that depends on alternative fuel type, quality and composition.

Therefore, a unique solution for burner technology is difficult to identify. There are many efficient technologies available, and the user must decide according to maximum GHG emissions, fuel availability and cost, required clinker quality and quantity, energy efficiency and overall costs that would fit a specific case.

**Author Contributions:** Conceptualization, M.M.M.; methodology, M.M.M.; investigation, D.M.C. and T.N.; validation, D.M.C. and M.M.M.; writing—original draft preparation, T.N. and D.M.C.; writing—review and editing, M.M.M. and D.M.C.; supervision, M.M.M. All authors have read and agreed to the published version of the manuscript.

**Funding:** The authors gratefully acknowledge the support of CERENA, through the Strategic Project FCT-UID/ECI/04028/2019 and to P2020 Clean Cement Line project (LISBOA-01-0247-FEDER-027500).

**Conflicts of Interest:** The authors declare no conflict of interest.

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
