# Peer review of "Modern Kiln Burner Technology in the Current Energy Climate: Pushing the Limits of Alternative Fuel Substitution"

_fire, doi:10.3390/fire6020074_

Round 1
Reviewer 1 Report
The presented review article is about kiln burner technology, but the part announced in the title about pushing the limits of alternative fuel substitution is difficult to trace (the mention of GHG is too little). The given overview is of good size and can be improved by considering the comments below. The evolution of burners is interesting, but it lacks the insights typical of a scientific paper (see notes below). Introductions must be more specific, more reflective of the specifics of the article, presented in their current form must be modified (reading only them gives the impression that it is not necessary to read the article itself, because the presented truths are already known).
List of remarks:
1. Lines 28-35. The section 0 must be removed from the paper. It is recommended for authors.
2. From the line 42, all abbreviations of pollutions should be written in correct form (for ex. In CO2 the 2 it should be written in subscript case).
3. Line 111. Cement manufacturers declare that it makes no difference to them what kind of fuel to burn, so the option with waste has much greater potential than granular coal. In this case, it is a win-win, because you need to dispose of the waste and obtain energy that can be used in the production of cement. In this case, it should be mentioned that not all waste can be burned, it is necessary to separate biodegradable, hazardous waste, which has its own specific use. Another aspect exhaust cleaning systems must be adapted accordingly, I think these aspects should be mentioned in the review.
4. Another aspect that must be highlighted is the paper's scientific novelty. It must be more reasonable, because such energy resources have been used in cement production for quite a long time.
5. Line 230. Returning to biodegradable waste, it is difficult to justify burning such waste when it is an excellent source for producing other fuels (eg methane) and obtaining fertilizers (compost). Such energy sources for the cement industry are possible, but not environmentally rational (from a waste recycling perspective).
6. Line 497. The evolution of burners is reviewed, but what is the scientific value of this information, where is the comparative analysis of different generations of burners presented? This presentation of material is more typical of a textbook, but not of a scientific article. Researchers reading this paper should not only familiarize themselves with the generations of burners, but also get a comparison of their criteria and a vision of how they could develop further from the technological side, what technological limitations may appear, etc. It goes without saying that manufacturers do not really want to reveal the intricacies of various burners, but the general trends must be highlighted, not only by mentioning the use of alternative fuels and GHG emissions (Discussion section).
7. Line 513. It is practically impossible to prevent the formation of high-temperature NOx in cement support. Therefore, it is necessary to focus on their cleaning technologies. So I think it should be mentioned here.
8. Line 550. It should be emphasized that when using alternative fuels, the diameter of solid particles is smaller. Therefore, the amount of particles themselves increases, but their diameter decreases (PM decreases, but PN increases).
9. Lines 624-631. Authors' contributions must be indicated.
Author Response
Dear reviewers,
We are grateful for your consideration and valued feedback. We have made necessary revisions and highlighted them in the updated manuscript, incorporating the suggestions provided.
Reviewer 1
List of remarks:
- Lines 28-35. The section 0 must be removed from the paper. It is recommended for authors.
The authors have removed this section as per the instructions.
- From the line 42, all abbreviations of pollutions should be written in correct form (for ex. In CO2 the 2 it should be written in subscript case).
Thank you for the remark. This issue has been solved.
For example: “SO2 is generated from the sulphur compounds in the ores and the combusted fuel”
- Line 111. Cement manufacturers declare that it makes no difference to them what kind of fuel to burn, so the option with waste has much greater potential than granular coal. In this case, it is a win-win, because you need to dispose of the waste and obtain energy that can be used in the production of cement. In this case, it should be mentioned that not all waste can be burned, it is necessary to separate biodegradable, hazardous waste, which has its own specific use. Another aspect exhaust cleaning systems must be adapted accordingly, I think these aspects should be mentioned in the review.
A brief comment on gas cleaning systems has been included. It should be stressed that although not all waste can be used, the term Waste Fuel (and each specific example) refers to a waste fraction that has been previously processed to remove the non-processable materials and isolate a combustible fraction with improved heating value, as is written in the section
“Waste fuels are derived from waste fractions (i.e. municipal solid waste) which are purified and separated in an attempt to isolate a fraction of the material possessing lower moisture content and higher calorific value to be used as fuel”
- Another aspect that must be highlighted is the paper's scientific novelty. It must be more reasonable, because such energy resources have been used in cement production for quite a long time.
This issue has been addressed. A paragraph has been inserted in the introduction section stating
“This work focuses on presenting and comparing modern kiln burner technologies in the current energy climate, as pertaining to maximizing alternative fuel processing and minimizing environmental impact. To the best of the authors’ knowledge, there are no recent reviews published on the subject.”
- Line 230. Returning to biodegradable waste, it is difficult to justify burning such waste when it is an excellent source for producing other fuels (eg methane) and obtaining fertilizers (compost). Such energy sources for the cement industry are possible, but not environmentally rational (from a waste recycling perspective).
The authors feel this discussion is beyond the scope of the paper, since it pertains to a broader waste-to-energy subject.
- Line 497. The evolution of burners is reviewed, but what is the scientific value of this information, where is the comparative analysis of different generations of burners presented? This presentation of material is more typical of a textbook, but not of a scientific article. Researchers reading this paper should not only familiarize themselves with the generations of burners, but also get a comparison of their criteria and a vision of how they could develop further from the technological side, what technological limitations may appear, etc. It goes without saying that manufacturers do not really want to reveal the intricacies of various burners, but the general trends must be highlighted, not only by mentioning the use of alternative fuels and GHG emissions (Discussion section).
This issue has been addressed by introducing a comment in the beginning of the discussion section. It should be noted that the first and second generations of burners are not recent technologies in the slightest and do not contemplate the processing of alternative fuels. As the name of the manuscript implies, this discussion is focused on modern kiln burner technologies and the authors feel that a detailed comparison of different burner generations does not add value to the overall publication.
“Kiln burner generations are defined by different operational paradigms. The first gen-eration of burners were used in wet or semi-wet production processes, where water was used to homogenize the raw material. This meant that a significant portion of the fuels were burned to evaporate water, rather than simply processing the raw material.
The second generation of burners was introduced in dry processes with shorter kilns. They focused on improving fuel injection (multi-channel injection) and primary air insertion to provide better flame control, employing mostly oil and coal as fuels.
Modern kiln burner technology, however, is focused on processing alternative fuels and maximizing their insertion, as well as improving the energy efficiency of the combustion process.”
- Line 513. It is practically impossible to prevent the formation of high-temperature NOx in cement support. Therefore, it is necessary to focus on their cleaning technologies. So I think it should be mentioned here.
This comment has been addressed with a paragraph.
“The available technical specifications for FCT’s Turbu-jet are as follows:
o Burner capacity: up to 160 MW;
o Solid, liquid and gaseous fuel support;
o Primary air percentage: approximately 11% (based on a case study);
o Can achieve alternative fuel substitution rates of 70% (based on a case study);”
- Line 550. It should be emphasized that when using alternative fuels, the diameter of solid particles is smaller. Therefore, the amount of particles themselves increases, but their diameter decreases (PM decreases, but PN increases).
In fact this is not usually the case, at least in cement production. The diameter of solid alternative fuels particles are one to two orders of magnitude higher than for coal/petcoke particles. This is due to the coal/petcoke milling operation before being fed to the burner. Solid alternative fuels such as RDF come prepared from the supplier and current comminution technologies are not as effective in reducing particle size for this type of fuels.
- Lines 624-631. Authors' contributions must be indicated.
Autor contribution has been listed.
“Author Contributions: Conceptualization, Margarida Mateus; methodology, Margarida Mateus; investigation, Duarte Cecílio and Teresa Neuparth; validation, Duarte Cecílio and Margarida Mateus; writing – original draft preparation, Teresa Neuparth and Duarte Cecílio; writing – review and editing, Margarida Mateus and Duarte Cecílio; supervision, Margarida Mateus.”
Thank you,
Margarida Mateus

Reviewer 2 Report
This review paper is proposed for publication aiming to describe the current technology availability of kiln burners in order to address the goal of fossil fuel substitution, and ultimately GHG reduction in the cement manufacturing industry.
This topic fits the scope of the journal and, at least in my opinion, it is an interesting introduction to the burning technology applied to kilns. Even for not specialized readers the substitution of fossil fuels with alternative ones appears challenging. RDF is abundant but probably its efficient use in kilns still hide few issues that furnace operators and burners providers do not fully disclose or do not have real experience with. Authors recognize that burners providers do not disclose full information, thus making a comparison between each model described in the paper become difficult.
With all that said, I have some recommendations to authors:
1) Please clean the template from the text which is not pertinent to the paper: section 0 for example, should be removed, and also the disclaimers at the end of the template.
2) Please double check the English wording since there are few mistakes in the text.
3) Section 1 is too long and should be reduced. Authors repeat several times the motivation and the importance of GHG reduction, (which is obviously the main driver for fuel substitution), but text of this section should be rewritten to avoid this repetition.
4) In section 1, Authors should add few numerical data about the thresholds set for kilns about emissions, pollutants, VOC, NOx etc. with the aim to allow readers to compare the current state of the art with the challenges coming from fuel substitution.
5) Fig.1 should be improved (image quality is low) and better adhere to the description of lines 114 and ff.
6) L226 and ff. Authors list a number of alternative fuels which should reduce the GHG emissions. This statement should be better explained and expanded because GHG reduction comes from a LCA analysis rather than a lower emission factor associated to those alternative fuels. Meat and bones, TDF, RDF, for example, play a different role in reducing GHG emissions in kilns since f.i. used tires are waste stream with energy potential, but not renewable sources like biomass.
7) Section 4. Despite providers of the described burners might not fully disclose their characteristics, for sure they make available technical data sheets. Those data sheets should be included in the paper, either following each description, or as a comparative table.
8) Similarly, in the discussion, numerical data about the environmental performance of the burners with conventional or alternative fuels should be disclosed. Authors are presenting a review paper and numerical data should be present.
Author Response
Dear Reviewer,
We extend our appreciation for your time and comments.
Your feedback has been informative and we have made the necessary changes to the manuscript. The revisions have been done in accordance with the suggestions.
- Please clean the template from the text which is not pertinent to the paper: section 0 for example, should be removed, and also the disclaimers at the end of the template.
The authors have removed the relevant section.
- Please double check the English wording since there are few mistakes in the text.
The document has been reviewed and detected mistakes were corrected.
3) Section 1 is too long and should be reduced. Authors repeat several times the motivation and the importance of GHG reduction, (which is obviously the main driver for fuel substitution), but text of this section should be rewritten to avoid this repetition.
Section 1 has been reduced and the motivation redundancies have been addressed.
4) In section 1, Authors should add few numerical data about the thresholds set for kilns about emissions, pollutants, VOC, NOx etc. with the aim to allow readers to compare the current state of the art with the challenges coming from fuel substitution.
Emissions limit values have been introduced, particularly for SO2, NOx and particulate matter in the cement industry. It is somewhat difficult to pinpoint exact values because the limits actually depend on the clinker production capacity of the kilns and the type of technology but exact values were introduced as it was most accurately possible to determine.
5) Fig.1 should be improved (image quality is low) and better adhere to the description of lines 114 and ff.
Fig 1 quality has been improved. (Please see the attachment)
6) L226 and ff. Authors list a number of alternative fuels which should reduce the GHG emissions. This statement should be better explained and expanded because GHG reduction comes from a LCA analysis rather than a lower emission factor associated to those alternative fuels. Meat and bones, TDF, RDF, for example, play a different role in reducing GHG emissions in kilns since f.i. used tires are waste stream with energy potential, but not renewable sources like biomass.
The comment has been addressed and an explanation was included.
“The CO2 reduction mechanism from using alternative fuels depends on the nature of the fuel itself. Biomass based fuels are renewable energy sources and waste derived fuels (RDF, TDF, meat and bone meal, etc.) are waste stream resources with significant energy content, thus reducing CO2 emissions by avoiding the typical end-of-life dis-posing mechanisms and physical-chemical process that result in higher CO2 emissions.”
7) Section 4. Despite providers of the described burners might not fully disclose their characteristics, for sure they make available technical data sheets. Those data sheets should be included in the paper, either following each description, or as a comparative table.
The comment has been addressed as much as possible. Many suppliers do not have enough technical information available, and some was obtained from case studies. Direct comparison is sometimes impossible.
The primary challenges in addressing this case include the following two factors:
Kiln size and capacity: Ensuring that the burner can provide the necessary amount of heat to maintain the desired temperature in the kiln.
Kiln configuration: Ensuring that the burner is compatible with the specific design and layout of the kiln.
8) Similarly, in the discussion, numerical data about the environmental performance of the burners with conventional or alternative fuels should be disclosed. Authors are presenting a review paper and numerical data should be present.
This comment has been addressed to the best possible ability of the authors. Burner performance is extremely dependent on the capacity of the corresponding rotary kiln, which makes direct comparisons sometimes impossible.
Thank you,
Margarida Mateus

Round 2
Reviewer 2 Report
Authors have addressed most of the comments. By reading the revised version is now clear that environmental performances are mostly dependent on the kiln geometry, rather than burner type. Still, information provided for reviewed burner types is scarce, but at least is an attempts to disclose such information to the public.